# Propensity Score-Matched Analysis of Neoadjuvant vs. Adjuvant Therapy in Renal Cell Carcinoma

**DOI:** 10.3390/cancers17213481

**Published:** 2025-10-29

**Authors:** Cesare Saitta, Giacomo Musso, Giuseppe Garofano, Hajime Tanaka, Dattatraya Patil, Margaret F. Meagher, Srinivas Vourganti, Edward Cherullo, Michael Liss, Marco Paciotti, Giovanni Lughezzani, Nicolò M. Buffi, Viraj Master, Yasuhisa Fujii, Rana R. McKay, Ithaar H. Derweesh

**Affiliations:** 1Department of Urology, UC San Diego Health System, San Diego, CA 92121, USA; saittacesare@icloud.com (C.S.); musso.giacomo@hsr.it (G.M.); giuseppe.garofano@humanitas.it (G.G.); mfmeaghe@health.ucsd.edu (M.F.M.); liss@health.ucsd.edu (M.L.); 2Department of Urology, IRCCS Humanitas Clinical and Research Hospital, 20089 Rozzano, Italy; marco.paciotti@humanitas.it (M.P.); giovanni.lughezzani@hunimed.eu (G.L.); nicolo.buffi@hunimed.eu (N.M.B.); 3Department of Biomedical Sciences, Humanitas University, Pieve Emanuele, 20072 Milan, Italy; 4Department of Urology, Tokyo Medical and Dental University, Tokyo 113-8510, Japan; hjtauro@tmd.ac.jp (H.T.); y-fujii.uro@tmd.ac.jp (Y.F.); 5Department of Urology, Emory Medical Center, Atlanta, GA 30322, USA; dattatraya.patil@emory.edu (D.P.); virajmaster@gmail.com (V.M.); 6Department of Urology, Rush University Medical Center, Chicago, IL 60612, USA; srinivas_vourganti@rush.edu (S.V.); edward_cherullo@rush.edu (E.C.); 7Department of Internal Medicine, Division of Hematology and Medical Oncology, UC San Diego School of Medicine, San Diego, CA 92093, USA; rmckay@health.ucsd.edu

**Keywords:** adjuvant therapy, carcinoma, renal cell, nephrectomy, neoadjuvant, propensity score match, survival outcomes

## Abstract

Patients with high-risk localized kidney cancer often receive systemic therapies either before or after surgery, but it is unclear which approach leads to better outcomes. In this study, we compared survival and recurrence rates in patients treated with neoadjuvant therapy versus adjuvant therapy. We used a propensity score matching model to fairly compare the two groups. Our results suggest that receiving treatment before surgery may reduce the risk of death and cancer recurrence, especially when using targeted therapy or immunotherapy. These findings highlight the need for future clinical trials to determine the best timing for systemic treatment in patients with aggressive kidney cancer.

## 1. Introduction

Renal Cell Carcinoma (RCC) constitutes 4% of tumors, accounting for 14,890 deaths in 2023 [1]. Because of earlier detection, there has been a notable uptick in the incidence of RCC, resulting in more localized and earlier-stage tumors [2]. Nonetheless, 40% of patients exhibit locally advanced disease at time of diagnosis or experience disease relapse after surgery [3]. Initial FDA approval of sunitinib and pembrolizumab as adjuvant therapies (ATs) has marked a significant shift in the management of high-risk localized RCC (HLR-RCC) [4,5]. On another note, neoadjuvant systemic therapy (NT) has been utilized investigatively to enhance the probability of complete surgical resection or to enable nephron-sparing surgery (NSS) in patients with imperative indication in this respect [6,7]. To the best of our knowledge, a comparison between oncological outcomes of NT and AT, as well as between immunotherapy (IO) and targeted molecular therapy (TMT), has not yet been conducted. Herein, we sought to evaluate oncological outcomes of surgically treated patients who have undergone either AT or NT in nm-RCC utilizing a propensity score-matched (PSM) model and to further elucidate the impact the different therapies provide in both settings.

## 2. Methods

### 2.1. Population

Institutional review board approval was obtained at participating centers (University of California San Diego, Rush University, Emory University and Tokyo Medical and Dental University). We conducted a multicenter retrospective analysis of surgically treated patients with T1-T4M0 HRL-RCC who underwent AT or NT from April 2006 to July 2022. Patients with clinical metastases at the time of surgery were excluded. All patients were staged with a CT scan of the chest, abdomen and pelvis. AT was defined as systemic therapy given postoperatively in the absence of documented metastases within six months after surgery. The receipt of AT was dictated according to NCCN guidelines [8] or as part of previous trials [9,10,11]. NT was given in the setting of locally advanced disease and clinical trials, performed to increase the likelihood of complete surgical resection in patients with imperative indications for NSS [7,12]. The decision on the type of surgery [partial nephrectomy (PN) or radical nephrectomy (RN)] was made by individual surgeons within the context of shared decision making. Postoperative follow-up was conducted according to institutional protocols and relevant guidelines [8,13].

### 2.2. Data

The data collected were age, sex, BMI, diabetes, hypertension, Charlson comorbidity index (CCI) score [14]], preoperative estimated glomerular filtrate rate (eGFR), surgical approach (RN vs. PN), Clavien–Dindo postoperative complication [15], chemotherapy received and timing of therapy. Pathological outcomes included TNM stage [16], tumor size, histology, grade [17], necrosis and presence of positive surgical margin (SM). The outcomes analyzed were All-Cause Mortality (ACM)/Overall Survival (OS), Cancer-Specific Mortality/Survival (CSM/CSS) and recurrence/Progression-Free Survival (PFS), which were computed from the day of surgery to the date of the event or last follow-up.

### 2.3. Statistical Analysis

The primary outcome was ACM; the secondary outcomes were CSM and recurrence. A PSM model was conducted with a caliper width of 0.1, including age, CCI, tumor size, necrosis, grade, stage, surgical approach and systemic therapy received. A nearest neighbor matching algorithm in a 1:2 ratio was used. Continuous variables were compared with the Wilcoxon rank sum test; categorical variables were analyzed using the Pearson chi-square test. Multivariable analysis (MVA) via Cox regression and Kaplan–Meier analysis (KMA) were conducted to elucidate predictors of outcomes and survival estimates. Subset MVA was conducted using an interaction term between chemo setting (AT vs. NT) and type of chemotherapy (TMT vs. IO) to elucidate how the effect of the chemo setting might vary depending on different regimens. Data were analyzed using Stata18.5/SE package software (StataCorp, Houston, TX, USA).

## 3. Results

### 3.1. Descriptive Analysis

Demographic characteristics are summarized in Table 1a,b. Overall, 347 patients met the inclusion criteria [AT *n* = 257 (132 TMT vs. 125 IO); NT *n* = 90 (61 TMT vs. 29 immunotherapy]; the median follow-up was 44 months (IQR 20–74). Patients in the neoadjuvant group were younger [median age 60 (IQR 53–69) vs. 65.5 (IQR 56–70), *p* = 0.007], more likely to have a CCI < 5 [77 (30%) vs. 36 (40%), *p* = 0.001], had a greater distribution of TMT [61 (67.8%) vs. 132 (51.4%), *p* = 0.007], a greater distribution of PN [19 (21.1%) vs. 31 (12.1%), *p* = 0.035] and fewer instances of tumor necrosis [54 (60%) vs. 122 (47.5%), *p* = 0.041].

### 3.2. Survival Analysis

MVA revealed hypertension (HR 1.52, *p* = 0.044), positive SM (HR 1.94, *p* = 0.046) and AT (HR 1.97, *p* = 0.007) as independent risk factors associated with an increased risk of ACM, while immunotherapy (HR 0.60, *p* = 0.011) was associated with a decreased risk (Table 2a). Subset MVA for ACM (Table 3a) revealed neoadjuvant TMT (HR 0.49, *p* = 0.016), neoadjuvant IO (HR 0.32, *p* = 0.016) and adjuvant IO (HR 0.59, *p* = 0.015) as associated with a decreased risk of ACM. MVA revealed hypertension (HR 1.88, *p* = 0.011), Stages III-IV vs. I-II (HR 1.76, *p* = 0.047) and AT (HR 2.37, *p* = 0.007) as risk factors for CSM, while immunotherapy (HR 0.59, *p* = 0.029) was associated with a decreased risk (Table 2b). Subset MVA for CSM (Table 3b) revealed neoadjuvant TMT (HR 0.47, *p* = 0.036) and neoadjuvant IO (HR 0.18, *p* = 0.017) as associated with decreased CSM. MVA revealed increasing tumor size (HR 1.06, *p* = 0.016), positive SM (HR 3.10, *p* < 0.001) and AT (HR 1.64, *p* = 0.02) as associated with recurrence (Table 2c). Subset MVA (Table 3c) revealed that receipt of neoadjuvant IO, TMT and adjuvant IO was not associated with a decreased risk of recurrence (*p* = 0.068, *p* = 0.184 and *p* = 0.818, respectively). KMA comparing neoadjuvant vs. adjuvant patients demonstrated 5-year OS of 79.7% vs. 61% (*p* = 0.009, Figure 1a), 5-year CSS of 87.6% vs. 72.8% (*p* = 0.01, Figure 1b) and 5-year PFS of 74.7% vs. 60% (*p* = 0.14, Figure 1c), respectively.

## 4. Discussion

We present the first multicenter analysis to explore the impact of timing and type of therapy on oncological outcomes in HRL-RCC. Our analysis revealed that receipt of NT was associated to a lower risk of ACM, CSM and recurrence. Our results suggest that NT may be of benefit in select patients with high-risk localized disease and that this benefit may be provided by TKI as well as IO agents. Conversely, while single-agent TKI provided similar benefits to IO-based protocols in the NT setting, IO was superior to TKI in the AT setting. In addition to supporting and validating IO clinical trials, which suggest superior outcomes in AT as opposed to TKI-based protocols in HRL-RCC, our findings call for the consideration of the investigation of NT vs. AT protocols in HRL-RCC in the context of a randomized clinical trial.

Adjuvant pembrolizumab has led to a paradigm shift in the management of HRL-RCC [6]. The KEYNOTE-564 trial [5] elucidated the impact of adjuvant pembrolizumab in 994 high-risk ccRCC patients. Regarding KMA, the authors noted that when comparing pembrolizumab vs. placebo, the 2-year DSF 77.3% vs. 68.1% (HR for recurrence/death 0.68, *p* = 0.02), while the 2-year OS was 96.6% vs. 93.5%. Nevertheless Choueiri et al. [18] revealed a notable improvement in OS in the group receiving pembrolizumab compared to the placebo after 5 years (HR 0.62, *p* = 0.0024). The S-TRAC trial [4] included 615 high-risk ccRCC patients who were randomized to receive sunitinib vs. placebo. After a median follow-up of 5.4 years, the authors noted that DFS was longer in the sunitinib group (6.8 vs. 5.6 years (HR 0.76, *p* = 0.03)). However, the occurrence of grade ≥ 3 complications in the sunitinib arm was not negligible (60.5% vs. 19.4%).

However, several studies have yielded negative results, and AT faces psychological hurdles of patient acceptance [19]. The ASSURE trial [10] corroborated the impact of adjuvant TMT in 1943 HRL-RCC patients with a median follow-up of 5.8 years. Patients were randomized in a 1:1:1 ratio to receive sunitinib, sorafenib or placebo, and no statistically significant differences in terms of DFS were present (HR 1.02, 97.5% CI 0.85–1.23). However, both treatment arms experienced over 55% adverse events. Similarly, the ATLAS trial [9], which compared axitinib vs. placebo, concluded without significant differences in terms of DFS. The PROTECT trial [11] compared pazopanib vs. placebo in 1538 patients, and like the ASSURE trial, 60% of patients in the treatment arm experienced adverse events. The Checkmate 914 trial [20] enrolled 816 patients who were randomized to either undergo AT with nivolumab plus ipilimumab or placebo. The median DSF was not ascertainable in the treatment group and was 50.7 months in the placebo cohort (HR 0.92, *p* = 0.53). Grade ≥ 3 adverse events were noted in 155 patients in the treatment group. The IMmotion010 trial [21] enrolled 778 high-risk RCC patients and compared atezolizumab vs. placebo, noting no differences in terms of DSF (*p* = 0.50). These findings suggest that the benefit of AT has been inconsistently observed with TMT, and while IO has shown superior outcomes with respect to two published trials [20,21], these findings were not significant using an alpha level of 0.05; a larger trial has demonstrated benefits with respect to DSF [5] and OS [18]. Our findings suggest that adjuvant IO is associated with a decreased risk of ACM (HR 0.59, *p* = 0.015), thus building upon prior work, and represent the first direct comparison of the two modalities.

The rationale of NT is an emerging investigational strategy to facilitate complete surgical resection in HLR-RCC characterized by extensive locoregional disease or NSS in complex renal tumors with imperative indications for nephron preservation [6,7,22]. Field et al. [12] conducted a study on 53 patients contrasting neoadjuvant sunitinib (*n* = 19) vs. surgery (*n* = 34) and noted that NT was associated with improved CSS (OR = 3.28; *p* = 0.021; KMA: 72 vs. 38 months, *p* = 0.023). Lane et al. [23] used sunitinib in 72 patients with a median tumor size of 7.2 cm (IQR 5.3–8.7 cm) and noted a reduction in tumor size to 5.3 cm (IQR: 4.1–7.5 cm), 59% reduction in RENAL score and 14% grade ≥ 3 toxicity. Hakimi et al. [7] noted that axitinib led to a decrease in tumor size (7.5–6.2 cm, *p* < 0.001) and RENAL score (11–10, *p* < 0.001) in 27 patients with imperative indication for PN, which was successfully performed in 74.0% of cases, with 96.2% achieving negative SM. These findings suggest that neoadjuvant TMT contributes to a moderate reduction in tumor size, enabling surgical resection in locoregional disease and for complex PN in patients with imperative indication; nonetheless, there is a gap in our knowledge regarding how localized RCC patients who receive NT perform oncologically, whether as a single-arm cohort or as a comparative study. Furthermore, surgical resection after NT remained intricate, demanding specialized surgical skills, and was associated with a non-negligible frequency of post-surgical complications [6].

Neoadjuvant IO in localized and locally advanced RCC is a burgeoning field of investigation [6,24]. Gorin et al. [25] elucidated the impact of neoadjuvant nivolumab in 17 nm-RCC patients [median tumor size 7.9 cm (6.7–8.5); stage breakdown: 6 (35.3%) cT2a, 1 (5.9%) cT2b, 7 (41.2%) cT3a and 3 (17.6%) cT3b] and noted that 53.3% experienced shrinkage in tumor size, with no Clavien grade ≥ 3 postoperative complications. Hakimi et al. [26] noted that neoadjuvant IO was correlated with significant decreases in tumor size [median change 1.9 cm (IQR −1.8, 5.6), *p* < 0.001), tumor thrombus dimension [median change 3.5 cm (IQR −1.2, 4.7), *p* = 0.02] and tumor complexity [RENAL score: pre-treatment: 9.2 (7.5, 10.9) vs. post-treatment 8.4 (6.5, 10.3), *p* < 0.001]. They also noted downstaging in 25 patients, which was associated with improved PFS (HR 5.15, *p* = 0.02). The Neoavax trial [27] assessed neoadjuvant avelumab/axitinib in 40 HRL-RCC patients and noted a partial response in 12 patients. In our retrospective cohort of 90 patients treated with neoadjuvant therapy, we observed that oncologic outcomes were comparable between IO and TMT, with no significant differences in ACM (*p* = 0.217), CSM (*p* = 0.247), recurrence (*p* = 0.876) or post-surgical complications (TMT 7% vs. IO 1%, *p* = 0.21).

Interestingly, these findings stand in contrast to the recently published PROSPER EA8143 trial [28], a randomized phase 3 study that evaluated perioperative nivolumab in patients with non-metastatic RCC. In that trial, 819 patients were randomized to receive either neoadjuvant and adjuvant nivolumab or surgery followed by surveillance. The trial was stopped early due to futility at interim analysis, with no significant improvement in recurrence-free survival observed (HR 0.94, 95% CI 0.74–1.21, *p* = 0.32). These findings raise important questions about the clinical benefit of perioperative IO in unselected patients. Cumulatively, the current body of evidence, including PROSPER and earlier phase 1–2 experiences, highlight that while neoadjuvant IO is generally well tolerated and may offer cytoreductive benefits, its impact on long-term oncological outcomes remains uncertain. However, our real-world propensity score-matched analysis suggests that NT, including IO, may confer survival advantages over AT, especially in selected high-risk patients. Notably, we observed that receipt of AT was independently associated with increased risk of ACM, CSM and recurrence, whereas IO, when given in the neoadjuvant setting, was associated with decreased risk across multiple endpoints. On the other hand, the negative results of the PROSPER trial may, in part, reflect pragmatic design limitations, such as a short preoperative treatment duration, the inclusion of patients with lower-risk features and absence of translational endpoints. Conversely, our cohort was restricted to high-risk, surgically treated, non-metastatic patients, and treatment duration was not constrained by trial protocol, potentially allowing for more meaningful immunologic priming. These differences may help explain the more favorable oncological outcomes observed in our neoadjuvant group.

Indeed, the question of the impact of NT on survival outcomes has not been well studied and has not been compared with a matched group that has done AT. Our analysis provides the first suggestion that in addition to facilitating surgical resection, NT may be associated with improved survival outcomes in HRL-RCC when compared to AT. Indeed, we noted that receipt of AT was associated with an increased risk of ACM and CSM (HR = 1.97, *p* = 0.007; HR = 2.37, *p* = 0.007; HR 1.64, *p* = 0.02, respectively). Additionally, regarding KMA, when comparing neoadjuvant vs. adjuvant therapy, we noticed increased OS (79.7% vs. 61%, *p* = 0.009) and CSS (87.6% vs. 72.8%, *p* = 0.01). Our findings as such call for the consideration of prospective investigations of the utility of presurgical NT not just as a tool to facilitate surgical resection but as a strategy to reduce oncological risk in HRL-RCC.

Our findings must be interpreted with caution given the retrospective study design and the impact of selection, recall and historical bias, which may still be present despite us relying on a PSM model. Firstly, where there is a significant overlap in clinical staging between the populations, which forms the basis of the hypothesis for our investigation, it nonetheless represents two distinctive subpopulations with differing indications for systemic therapy: the neoadjuvant population was one in which surgical resectability was questionable, while in the adjuvant population, complete resection was achieved, and receipt of therapy was driven by the desire to consolidate therapeutic goals. Secondly, the distinct timelines in the application of AT and NT present an ongoing concern and may also lead to bias driven by differences in exposure to the systemic agent. Thirdly, adjuvant IO has only been available in a non-investigational indication, since November 2021; consequently, the limited duration of follow-up associated with this relatively recent introduction may contribute to outcome bias. The study did not include a matched cohort of HRL-RCC patients who did not undergo AT or NT; however, as the field has pivoted to the paradigm of adjuvant therapy for HRL-RCC, the inclusion of such a cohort from a multicenter analysis of patients would have been historical and would have decreased the power of the PSM model. The study did not involve a centralized review of the pathological samples and reports and, thus, can lead to discrepancies and outcome bias. Moreover, our database lacks information about the effect of neoadjuvant treatment on decreasing tumor size, downstaging and other data, including nephrometry score, side effects and the location of positive margins, as such does not allow for qualitatively and quantitatively statements to be made. While pathological response has emerged as a robust surrogate endpoint in other solid tumors, such as melanoma and bladder cancer, no standardized criteria exist for RCC. Detailed information on treatment duration was not available, which may have introduced variability between regimens and represents a limitation of the study. Our study, although limited by retrospective design and lack of pathological response assessment, supports the need for integrated translational endpoints such as major pathological response (MPR) or immune infiltration markers in future prospective trials [29]. Our findings may not be generalizable to centers without substantial experience in adjuvant or neoadjuvant protocols. Furthermore, we acknowledge that the inclusion of different TKI and IO regimens may be a potential source of bias. Finally, due to limited sample sizes within specific treatment subgroups, we were unable to assess differences in efficacy between individual IO or TKI agents. Nonetheless, our findings are bolstered by a substantial number of patients who underwent NT and AT and suggest that receipt of NT may be associated with improved oncological survival outcomes compared to AT, calling for the consideration of a better understanding of these findings in the context of a clinical trial.

## 5. Conclusions

Our findings suggest that receipt of NT was associated with superior survival outcomes and that adjuvant IO was associated with improved OS when compared to TMT. Our findings are hypothesis generating and call for further investigations regarding oncological outcomes and the utility of neoadjuvant IO and TMT therapy for HRL-RCC.

## Figures and Tables

**Figure 1 cancers-17-03481-f001:**
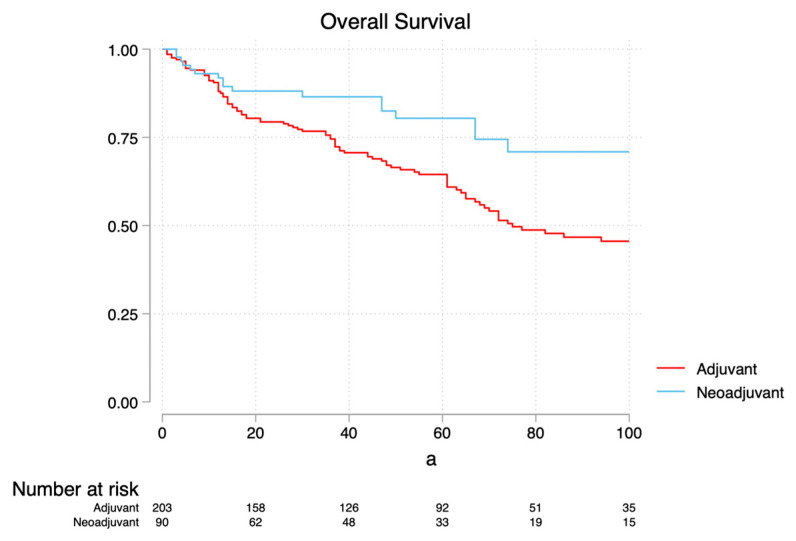
Kaplan–Meier Analysis for Overall Survival (**a**), Cancer Specific Survival (**b**) and Progression Free Survival (**c**), comparing adjuvant and neoadjuvant settings.

**Table 1 cancers-17-03481-t001:** (a) Systemic therapy breakdown before and after PSM. (b) Descriptive analysis.

**(a)**									
**Before PSM**				**After PSM**					
**Neoadjuvant Breakdown**	** *n* **	**%**	**Cumulative**	**Neoadjuvant Breakdown**		** *n* **	**%**	**Cumulative**	
Axitinib	29	32.22	32.22	Axitinib		29	32.22	32.22	
Nivolumab	29	32.22	64.44	Nivolumab		29	32.22	64.44	
Sunitinib	32	35.56	100	Sunitinib		32	35.56	100	
Total	90	100		Total		90	100		
**Before PSM**				**After PSM**					
**Adjuvant Breakdown**	** *n* **	**%**	**Cumulative**	**Adjuvant Breakdown**		** *n* **	**%**	**Cumulative**	
Atezolizumab	4	1.56	1.56	Atezolizumab		4	1.81	1.81	
Axitinib	13	5.06	6.61	Axitinib		12	5.43	7.24	
Pazopanib	35	13.62	20.23	Pazopanib		35	15.84	23.08	
Pembrolizumab	121	47.08	67.32	Pembrolizumab		90	40.72	63.8	
Sunitinib	84	32.68	100	Sunitinib		80	36.2	100	
Total	257	100		Total		221	100		
**(b)**									
**Before Propensity Score Matching**						**After Propensity Score Matching**			
			**Neoadjuvant**	**Adjuvant**	***p*-Value**	**Neoadjuvant**		**Adjuvant**	***p*-Value**
			***n* = 90**	***n* = 257**		***n* = 90**		***n* = 221**	
Age median (IQR)			65.5 (56–70)	60 (53–69)	0.007 §	65.5 (56–70)		60 (53–69)	0.037 §
Sex *n* (%)	Female		31 (34.4%)	76 (29.6%)	0.39 *	31 (34.4%)		56 (25.3%)	0.10 *
	Male		59 (65.6%)	181 (70.4%)		59 (65.6%)		165 (74.7%)	
Ethnicity *n* (%)	White		44 (48.9%)	173 (67.3%)	<0.001 *	44 (48.9%)		145 (65.6%)	<0.001 *
	Latin		11 (12.2%)	3 (1.2%)		11 (12.2%)		2 (0.9%)	
	Asian		29 (32.2%)	37 (14.4%)		29 (32.2%)		37 (16.7%)	
	Afro-American		3 (3.3%)	38 (14.8%)		3 (3.3%)		32 (14.5%)	
	Others		3 (3.3%)	6 (2.3%)		3 (3.3%)		5 (2.3%)	
BMI median (IQR)			25.9 (23.3–30.8)	27.9 (24–32.5)	0.085 §	25.9 (23.3–30.8)		27.6 (24–32.3)	0.088 §
Hypertension *n* (%)	No		49 (54.4%)	111 (43.2%)	0.065 *	49 (54.4%)		102 (46.2%)	0.18 *
	Yes		41 (45.6%)	146 (56.8%)		41 (45.6%)		119 (53.8%)	
Diabetes *n* (%)	No		70 (77.8%)	181 (70.4%)	0.18 *	70 (77.8%)		162 (73.3%)	0.41 *
	Yes		20 (22.2%)	76 (29.6%)		20 (22.2%)		59 (26.7%)	
CCI *n* (%)	<5		36 (40.0%)	77 (30.0%)	0.001 *	36 (40.0%)		76 (34.4%)	0.11 *
	5–8		39 (43.3%)	83 (32.3%)		39 (43.3%)		83 (37.6%)	
	≥9		15 (16.7%)	97 (37.7%)		15 (16.7%)		62 (28.1%)	
Preoperative eGFR median (IQR)			63.9 (52.7–71.7)	68 (56–82)	0.10 §	63.9 (52.7–71.7)		68 (56–82)	0.14 §
CKD-S3a *n* (%)	No		75 (83.3%)	230 (89.5%)	0.12 *	75 (83.3%)		198 (89.6%)	0.13 *
	Yes		15 (16.7%)	27 (10.5%)		15 (16.7%)		23 (10.4%)	
Systemic therapy received *n* (%)	TMT		61 (67.8%)	132 (51.4%)	0.007 *	61 (67.8%)		127 (57.5%)	0.092 *
	IO		29 (32.2%)	125 (48.6%)		29 (32.2%)		94 (42.5%)	
Systemic therapy breakdown	Atezolizumab		0 (0.0%)	4 (1.6%)	<0.001	0 (0.0%)		4 (1.8%)	<0.001
	Axitinib		29 (32.2%)	13 (5.1%)		29 (32.2%)		12 (5.4%)	
	Nivolumab		29 (32.2%)	0 (0.0%)		29 (32.2%)		0 (0.0%)	
	Pazopanib		0 (0.0%)	35 (13.6%)		0 (0.0%)		35 (15.8%)	
	Pembrolizumab		0 (0.0%)	121 (47.1%)		0 (0.0%)		90 (40.7%)	
	Sunitinib		32 (35.6%)	84 (32.7%)		32 (35.6%)		80 (36.2%)	
Tumor size, median cm (IQR)			6.05 (4.7–8.5)	6.8 (5.1–9.7)	0.069 §	6.05 (4.7–8.5)		6.7 (5.1–9.2)	0.16 §
Surgery type *n* (%)	PN		19 (21.1%)	31 (12.1%)	0.035 *	19 (21.1%)		29 (13.1%)	0.077 *
	RN		71 (78.9%)	226 (87.9%)		71 (78.9%)		192 (86.9%)	
Histology *n* (%)	non-ccRCC		29 (32.2%)	62 (24.1%)	0.13 *	29 (32.2%)		50 (22.6%)	0.078 *
	ccRCC		61 (67.8%)	195 (75.9%)		61 (67.8%)		171 (77.4%)	
Tumor necrosis *n* (%)	No		54 (60.0%)	122 (47.5%)	0.041 *	54 (60.0%)		109 (49.3%)	0.087 *
	Yes		36 (40.0%)	135 (52.5%)		36 (40.0%)		112 (50.7%)	
Tumor grade *n* (%)	Low grade		21 (23.3%)	90 (35.0%)	0.041 *	21 (23.3%)		66 (29.9%)	0.24 *
	High grade		69 (76.7%)	167 (65.0%)		69 (76.7%)		155 (70.1%)	
Stage AJCC *n* (%)	Stage I		28 (31.1%)	55 (21.4%)	0.088 *	28 (31.1%)		51 (23.1%)	0.22 *
	Stage II		8 (8.9%)	14 (5.4%)		8 (8.9%)		13 (5.9%)	
	Stage III		49 (54.4%)	178 (69.3%)		49 (54.4%)		148 (67.0%)	
	Stage IV		5 (5.6%)	10 (3.9%)		5 (5.6%)		9 (4.1%)	
Surgical margin *n* (%)	Negative		81 (90.0%)	240 (93.4%)	0.29 *	81 (90.0%)		205 (92.8%)	0.42 *
	Positive		9 (10.0%)	17 (6.6%)		9 (10.0%)		16 (7.2%)	
Complication *n* (%)	No		70 (77.8%)	195 (75.9%)	0.71 *	70 (77.8%)		173 (78.3%)	0.92 *
	Yes		20 (22.2%)	62 (24.1%)		20 (22.2%)		48 (21.7%)	
Clavien grade ≥ III *n* (%)	No		82 (91.1%)	257 (100.0%)	<0.001 *	82 (91.1%)		221 (100.0%)	<0.001 *
	Yes		8 (8.9%)	0 (0.0%)		8 (8.9%)		0 (0.0%)	
Status at last follow-up	Alive		70 (77.8%)	133 (51.8%)	<0.001 *	70 (77.8%)		115 (52.0%)	<0.001 *
	ACM		20 (22.2%)	124 (48.2%)		20 (22.2%)		106 (48.0%)	
Cause of death	Alive		70 (77.8%)	133 (51.8%)	<0.001 *	70 (77.8%)		115 (52.0%)	<0.001 *
	CSM		12 (13.3%)	84 (32.7%)		12 (13.3%)		72 (32.6%)	
	OCM		8 (8.9%)	40 (15.6%)		8 (8.9%)		34 (15.4%)	
Recurrence	No		61 (67.8%)	96 (37.4%)	<0.001 *	61 (67.8%)		84 (38.0%)	<0.001 *
	Yes		29 (32.2%)	161 (62.6%)		29 (32.2%)		137 (62.0%)	
Length of follow-up, median months (IQR)			40 (17–73)	44 (22–76)	0.33 §	40 (17–73)		46 (26–74)	0.27 §

§: Wilcoxon Rank Sum Test; *: Pearson’s Chi-Squared Test; IQR: Interquartile Range; BMI: Body Mass Index; CCI: Charlson Comorbidity Index; eGFR: Estimated Glomerular Filtrate Rate CKD-S3a: eGFR < 60 mL/min/1.73 m^2^; TMT: Target Molecular Therapy; IO: Immunotherapy; PN: Partial Nephrectomy; RN: Radical Nephrectomy; ccRCC: Clear-Cell RCC; AJCC: American Joint Committee Cancer; ACM: All-Cause Mortality; CSM: Cancer-Specific Mortality; OCM: Other-Cause Mortality.

**Table 2 cancers-17-03481-t002:** (a) MVA for ACM, (b) MVA for CSM and (c) MVA for recurrence.

**(a)**				
**Covariates**	**HR**	**95% CI**		***p*-Value**
Increasing Age	1.03	1.01	1.04	0.01
CKD ≥ 3a	0.72	0.44	1.17	0.179
Increasing BMI	1.02	0.99	1.04	0.198
HTN (Yes vs. No)	1.52	1.01	2.27	0.044
Increasing Tumor Size	1.02	0.96	1.08	0.51
RN vs. PN	1.17	0.66	2.09	0.593
Stages III–IV vs. I–II	1.38	0.86	2.21	0.18
Necrosis (Yes vs. No)	1.29	0.88	1.88	0.19
Positive Surgical Margin (Yes vs. No)	1.94	1.01	3.72	0.046
High Grade vs. Low Grade	0.94	0.61	1.45	0.789
Adjuvant vs. Neoadjuvant	1.97	1.20	3.24	0.007
IO vs. TKI	0.60	0.40	0.89	0.011
**(b)**				
**Covariates**	**HR**	**95% CI**		***p*-Value**
Increasing Age	1.03	1.00	1.05	0.018
HTN (Yes vs. No)	1.88	1.15	3.05	0.011
RN vs. PN	1.79	0.82	3.90	0.144
Stages III–IV vs. I–II	1.76	1.01	3.06	0.047
Positive Surgical Margin (Yes vs. No)	2.00	0.94	4.29	0.074
High Grade vs. Low Grade	1.57	0.90	2.75	0.111
Adjuvant vs. Neoadjuvant	2.37	1.27	4.43	0.007
IO vs. TKI	0.59	0.36	0.95	0.029
**(c)**				
**Covariates**	**HR**	**95% CI**		***p*-Value**
Increasing Age	1.03	1.01	1.05	<0.001
Increasing BMI	1.02	0.99	1.04	0.133
HTN (Yes vs. No)	1.14	0.80	1.62	0.458
Increasing Tumor Size	1.06	1.01	1.11	0.016
RN vs. PN	1.50	0.90	2.50	0.119
Stages III–IV vs. I–II	1.17	0.80	1.72	0.423
Necrosis (Yes vs. No)	1.09	0.78	1.53	0.619
Positive Surgical Margin (Yes vs. No)	3.10	1.77	5.43	<0.001
High Grade vs. Low Grade	1.10	0.75	1.61	0.632
Adjuvant vs. Neoadjuvant	1.64	1.08	2.50	0.02
IO vs. TKI	1.04	0.74	1.44	0.834

**Table 3 cancers-17-03481-t003:** (a) Multivariate survival analysis with chemotherapy and immunotherapy interaction term for ACM. (b) Multivariate survival analysis with chemotherapy and immunotherapy interaction term for CSM. (c) Multivariate survival analysis with chemotherapy and immunotherapy interaction term for recurrence.

**(a)**				
**Covariates**	**HR**	**95% CI**		***p*-Value**
Increasing Age	1.03	1.01	1.04	0.01
CKD ≥ 3a	0.72	0.44	1.17	0.18
Increasing BMI	1.02	0.99	1.04	0.2
HTN (Yes vs. No)	1.51	1.01	2.27	0.047
Increasing Tumor Size	1.02	0.96	1.08	0.499
RN vs. PN	1.16	0.65	2.09	0.608
Stages III–IV vs. I–II	1.38	0.86	2.22	0.179
Necrosis (Yes vs. No)	1.29	0.88	1.88	0.191
Positive Surgical Margin (Yes vs. No)	1.94	1.01	3.72	0.045
High Grade vs. Low Grade	0.94	0.61	1.45	0.786
Adjuvant vs. Neoadjuvant IO vs. TKI	Adjuvant TKI (Reference)			
Neoadjuvant TKI	0.49	0.28	0.88	0.016
Neoadjuvant IO	0.32	0.13	0.81	0.016
Adjuvant IO	0.59	0.38	0.90	0.015
**(b)**				
**Covariates**	**HR**	**95% CI**		***p*-Value**
Increasing Age	1.03	1.00	1.05	0.018
HTN (Yes vs. No)	1.90	1.17	3.10	0.01
RN vs. PN	1.82	0.83	3.98	0.133
Stages III–IV vs. I–II	1.74	1.00	3.04	0.049
Positive Surgical Margin (Yes vs. No)	2.00	0.93	4.27	0.074
High Grade vs. Low Grade	1.59	0.91	2.77	0.105
Adjuvant vs. Neoadjuvant IO vs. TKI	Adjuvant TKI (Reference)			
Neoadjuvant TKI	0.47	0.24	0.95	0.036
Neoadjuvant IO	0.18	0.04	0.73	0.017
Adjuvant IO	0.62	0.37	1.03	0.065
**(c)**				
**Covariates**	**HR**	**95% CI**		***p*-Value**
Increasing Age	1.03	1.01	1.05	0.001
Increasing BMI	1.02	0.99	1.04	0.134
HTN (Yes vs. No)	1.14	0.80	1.63	0.454
Increasing Tumor Size	1.06	1.01	1.11	0.017
RN vs. PN	1.50	0.90	2.51	0.119
Stages III–IV vs. I–II	1.17	0.80	1.72	0.423
Necrosis (Yes vs. No)	1.09	0.78	1.53	0.62
Positive Surgical Margin (Yes vs. No)	3.09	1.76	5.44	<0.001
High Grade vs. Low Grade	1.10	0.75	1.61	0.633
Adjuvant vs. Neoadjuvant IO vs. TKI	Adjuvant TKI (Reference)			
Neoadjuvant TKI	0.62	0.37	1.04	0.068
Neoadjuvant IO	0.62	0.30	1.26	0.184
Adjuvant IO	1.04	0.73	1.50	0.818

## Data Availability

The data supporting the findings of this study are not publicly available due to privacy and ethical restrictions related to the use of patient-level clinical data from multiple institutions.

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
