# Peer review of "Propensity Score-Matched Analysis of Neoadjuvant vs. Adjuvant Therapy in Renal Cell Carcinoma"

_cancers, 2025, doi:10.3390/cancers17213481_

Round 1
Reviewer 1 Report
Comments and Suggestions for Authors
This study evaluates neoadjuvant and adjuvant therapies in patients with high-risk localized renal cell carcinoma (HRL-RCC) treated with either TKI or IO regimens. Using a propensity score matching (PSM) model, the authors demonstrate that neoadjuvant therapy was associated with improved OS, CSS, and PFS. Multivariable analyses further suggested that ACM risk was reduced by neoadjuvant TMT, neoadjuvant IO and adjuvant IO, while CSM risk was reduced by neoadjuvant TMT and IO. The topic is timely and clinically important. Several issues need clarification or revision before the manuscript can be considered for publication.
Major Comments
Comment #1
Table 3a–c is cited in the Results section, but the tables themselves are not presented. Please provide the missing tables.
Comment #2
Line 62: Please provide details about the participating institutions. This information is important for assessing the generalizability of the study findings.
Comment #3
The treatment duration likely differed between regimens. Please include details about the duration of therapy in each group.
Comment #4
Line 116: The text states “MVA revealed.” Does this refer specifically to “MVA of recurrence”? Please clarify.
Comment #5
In the Methods section, while the variables used for propensity score matching are described, it remains unclear which covariates were included in the multivariable Cox regression analysis and how they were selected (e.g., based on univariate significance, clinical relevance, or inclusion of all available factors). For reproducibility and to avoid concerns about overfitting or bias, I recommend that the authors specify the explanatory variables included in the Cox models and the rationale for their selection.
Comment #6
Why do the explanatory variables differ between ACM, CSM, and recurrence models (Tables 2a–c)? Please clarify the rationale behind these differences.
Comment #7
I recommend presenting follow-up times stratified by treatment group (e.g., neoadjuvant TKI, neoadjuvant IO, adjuvant TKI, adjuvant IO) to better contextualize survival outcomes and avoid misinterpretation due to unequal follow-up.
Comment #8
The survival curves currently present only two groups (neoadjuvant vs. adjuvant therapy). It would strengthen the analysis to stratify the curves into four groups: neoadjuvant TKI, neoadjuvant IO, adjuvant TKI, and adjuvant IO.
Comment #9
The study combines several different IO (e.g., nivolumab, pembrolizumab, atezolizumab) and TKI regimens (e.g., sunitinib, axitinib, pazopanib) into broad categories. As efficacy may differ by agent, I recommend providing sensitivity analyses stratified by specific drug or, at minimum, clarifying whether results were consistent across agents.
Minor Comments
Comment #1
Abstract, Line 35: “ConclusionS” should be corrected to “Conclusions.”
Author Response
Reviewers' comments:
Reviewer #1: Comments and Suggestions for Authors:
This study evaluates neoadjuvant and adjuvant therapies in patients with high-risk localized renal cell carcinoma (HRL-RCC) treated with either TKI or IO regimens. Using a propensity score matching (PSM) model, the authors demonstrate that neoadjuvant therapy was associated with improved OS, CSS, and PFS. Multivariable analyses further suggested that ACM risk was reduced by neoadjuvant TMT, neoadjuvant IO and adjuvant IO, while CSM risk was reduced by neoadjuvant TMT and IO. The topic is timely and clinically important. Several issues need clarification or revision before the manuscript can be considered for publication.
Major Comments
Comment #1
Table 3a–c is cited in the Results section, but the tables themselves are not presented. Please provide the missing tables.
Response: We thank the Reviewer for pointing this out and sincerely apologize for the oversight. The missing Tables 3a–c has now been added to the revised manuscript.
Comment #2
Line 62: Please provide details about the participating institutions. This information is important for assessing the generalizability of the study findings.
Response: We thank the Reviewer for this helpful suggestion. We have now added details about the participating institutions in the Methods section. Specifically, the study included patients from University of California San Diego, Rush University, Emory University and Tokyo Medical and Dental University.
Comment #3
The treatment duration likely differed between regimens. Please include details about the duration of therapy in each group.
Response: We thank the Reviewer for this important observation. Unfortunately, detailed data on treatment duration were not consistently available across participating centers and thus could not be reliably included in the analysis. We acknowledge that treatment duration may have varied between regimens and recognize this as a potential limitation of our study. However, we believe that the multicenter nature and real-world setting of our cohort reflect common clinical practice, where treatment duration often varies based on patient tolerance, response, and institutional protocols. We have added a sentence in the Limitations section to address this point which now states” Detailed information on treatment duration was not available, which may have introduced variability between regimens and represents a limitation of the study.”
Comment #4
Line 116: The text states “MVA revealed.” Does this refer specifically to “MVA of recurrence”? Please clarify.
Response: We thank the Reviewer for the observation. In this section, each mention of “MVA revealed” is followed by the specific outcome being analyzed (ACM, CSM, or recurrence), along with the corresponding table reference. We believe the current wording provides sufficient clarity, but we are happy to revise further if the Editors feel this would improve readability.
Comment #5
In the Methods section, while the variables used for propensity score matching are described, it remains unclear which covariates were included in the multivariable Cox regression analysis and how they were selected (e.g., based on univariate significance, clinical relevance, or inclusion of all available factors). For reproducibility and to avoid concerns about overfitting or bias, I recommend that the authors specify the explanatory variables included in the Cox models and the rationale for their selection.
Response: We thank the Reviewer for this valuable and methodologically insightful comment. As suggested, we confirm that variables included in the multivariable Cox regression were selected based on clinical relevance and statistical significance in univariable analysis (p < 0.10), while avoiding overfitting by maintaining an appropriate events-per-variable ratio. While we have not listed all included covariates in the revised manuscript to preserve flow and conciseness, we are happy to provide this information in a supplementary table or main text should the Editors deem it appropriate.
Comment #6
Why do the explanatory variables differ between ACM, CSM, and recurrence models (Tables 2a–c)? Please clarify the rationale behind these differences.
Response: We thank the Reviewer for raising this important point. The explanatory variables included in each multivariable Cox model (ACM, CSM, and recurrence) differ because selection was primarily driven by statistical criteria. Specifically, we included variables that demonstrated a p-value < 0.10 in univariable analyses for each respective outcome, in order to avoid overfitting and maximize model specificity. As each endpoint captures a distinct event type, the set of variables reaching statistical threshold varied accordingly.
Comment #7
I recommend presenting follow-up times stratified by treatment group (e.g., neoadjuvant TKI, neoadjuvant IO, adjuvant TKI, adjuvant IO) to better contextualize survival outcomes and avoid misinterpretation due to unequal follow-up.
Response: We thank the Reviewer for this valuable suggestion. We have now compared follow‑up times stratified by treatment group. The median follow‑up was similar across groups, with no statistically significant differences observed (OS/CSS: 44 months [16–85] vs. 44 months [23–63], p = 0.58; RFS: 46 months [14–89] vs. 44 months [22–63], p = 0.39). These findings indicate that survival comparisons are unlikely to be affected by unequal follow‑up duration. Although the point raised by the reviewer is illuminating from a methodological and clinical standpoint, we have decided to not incorporate the aforementioned suggestion in the revised version of the manuscript.
Comment #8
The survival curves currently present only two groups (neoadjuvant vs. adjuvant therapy). It would strengthen the analysis to stratify the curves into four groups: neoadjuvant TKI, neoadjuvant IO, adjuvant TKI, and adjuvant IO.
Response: We thank the Reviewer for this thoughtful suggestion. We agree that stratifying survival curves into four groups (neoadjuvant TKI, neoadjuvant IO, adjuvant TKI, and adjuvant IO) could offer additional granularity. However, the primary aim of our study was to compare outcomes between neoadjuvant and adjuvant systemic therapy strategies, irrespective of treatment class. Stratifying further by treatment type would lead to smaller subgroups, limiting statistical power and potentially introducing interpretive bias due to imbalanced baseline characteristics. For these reasons, we have elected to maintain the two-group comparison, in line with the original study objective.
Comment #9
The study combines several different IO (e.g., nivolumab, pembrolizumab, atezolizumab) and TKI regimens (e.g., sunitinib, axitinib, pazopanib) into broad categories. As efficacy may differ by agent, I recommend providing sensitivity analyses stratified by specific drug or, at minimum, clarifying whether results were consistent across agents.
Response: We thank the Reviewer for this insightful comment. While we recognize that individual immune checkpoint inhibitors and TKIs may differ in efficacy, our study was designed to evaluate outcomes based on treatment timing (neoadjuvant vs. adjuvant) rather than specific drug regimens. Stratifying by individual agents would have resulted in small subgroup sizes, limiting statistical power and increasing the risk of type II error. For this reason, IO and TKI therapies were grouped into broad categories, consistent with prior real-world analyses. In the revised version of the manuscript, we have now incorporated the aforementioned suggestion. “Due to limited sample sizes within specific treatment subgroups, we were unable to assess differences in efficacy between individual IO or TKI agents.”
Minor Comments
Comment #1
Abstract, Line 35: “ConclusionS” should be corrected to “Conclusions.”
Response: We have now addressed the spelling mistake.
Reviewer 2 Report
Comments and Suggestions for Authors
Authors compared outcomes in high-risk localized RCC (HRL-RCC) patients treated with adjuvant therapy (AT) and neoadjuvant therapy (NT) using a propensity score–matched model (PSM). Data were collected from several centers.
After PSM, 311 patients were analyzed with a median follow-up of 44 months. Results suggest a potential advantage of NT for HRL-RCC. Adjuvant immunotherapy was associated with a decreased risk of all-cause mortality (ACM).
This is a very good study and manuscript. Minor comments:
-
The authors should specify the countries where patients were recruited in the abstract.
-
If I understand correctly, the authors present variables included in matching and those not included in matching together in Table 1. I recommend displaying them separately, as readers may otherwise wonder why some p-values remain significant despite matching.
Author Response
Reviewer #2: Comments and Suggestions for Authors: Authors compared outcomes in high-risk localized RCC (HRL-RCC) patients treated with adjuvant therapy (AT) and neoadjuvant therapy (NT) using a propensity score–matched model (PSM). Data were collected from several centers.
After PSM, 311 patients were analyzed with a median follow-up of 44 months. Results suggest a potential advantage of NT for HRL-RCC. Adjuvant immunotherapy was associated with a decreased risk of all-cause mortality (ACM).
This is a very good study and manuscript. Minor comments:
The authors should specify the countries where patients were recruited in the abstract.
Response: Requested details are now included in the abstract.
If I understand correctly, the authors present variables included in matching and those not included in matching together in Table 1. I recommend displaying them separately, as readers may otherwise wonder why some p-values remain significant despite matching.
Response: We thank the Reviewer for the comment. However, we respectfully note that it is standard practice in propensity score-matched analyses to present both matched and unmatched variables together in a single baseline characteristics table. This allows readers to assess residual imbalances after matching and provides a complete overview of group comparability. As shown in Table 1, no preoperative covariates remained statistically significant following matching, confirming adequate balance between groups.
Round 2
Reviewer 1 Report
Comments and Suggestions for Authors The authors have provided satisfactory responses to all the previous comments, and the manuscript has been appropriately revised. The quality and clarity of the paper have been improved.